# Factors influencing the delivery of telerehabilitation for stroke: A systematic review

Aoife Stephenson[1,2,3], Sarah Howes[2,3], Paul J. Murphy[4], Judith E. Deutsch[5], Maria Stokes[6,7], Katy Pedlow[2]*, Suzanne M. McDonough[1,2,3,8]

1 School of Physiotherapy, Royal College of Surgeons in Ireland, Dublin, Ireland, 2 Centre for Health and Rehabilitation Technologies, Ulster University, Newtownabbey, United Kingdom, 3 School of Health Sciences, Ulster University, Newtownabbey, United Kingdom, 4 RCSI Library, Royal College of Surgeons in Ireland, Dublin, Ireland, 5 Rivers Lab, Department of Rehabilitation and Movement Science, School of Health Professions, Rutgers University, New Brunswick, New Jersey, United States of America, 6 School of Health Sciences, University of Southampton, Southampton, United Kingdom, 7 Centre for Sport, Exercise and Osteoarthritis, Research Versus Arthritis, Chesterfield, United Kingdom, 8 School of Physiotherapy, University of Otago, Dunedin, New Zealand

☉ These authors contributed equally to this work.
* k.pedlow@ulster.ac.uk

**Data Availability Statement:** All relevant data are within the paper and its Supporting information files.

## Abstract

### Objective

Despite the available evidence regarding effectiveness of stroke telerehabilitation, there has been little focus on factors influencing its delivery or translation from the research setting into practice. There are complex challenges to embedding telerehabilitation into stroke services and generating transferable knowledge about scaling up and routinising this service model. This review aimed to explore factors influencing the delivery of stroke telerehabilitation interventions, including platforms, technical requirements, training, support, access, cost, usability and acceptability.

### Methods

MEDLINE, EMBASE, CINAHL, Web of Science and Cochrane Library and Central Registry of Clinical Trials were searched to identify full-text articles of randomised controlled trials (RCTs) and protocols for RCTs published since a Cochrane review on stroke telerehabilitation services. A narrative synthesis was conducted, providing a comprehensive description of the factors influencing stroke telerehabilitation intervention delivery.

### Results

Thirty-one studies and ten protocols of ongoing studies were included. Interventions were categorised as synchronous telerehabilitation (n = 9), asynchronous telerehabilitation (n = 11) and tele-support (n = 11). Telephone and videoconference were the most frequently used modes of delivery. Usability and acceptability with telerehabilitation were high across all platforms, although access issues and technical challenges may be potential barriers to

**Funding:** AS's time was co-funded by Royal College of Surgeons in Ireland (School of Physiotherapy) and the European Union's Horizon 2020 Research and Innovation Programme under grant agreement no. 687228, MAGIC PCP. The funders had no role in study design, data collection and analysis, decision to publish, or preparation of the manuscript.

**Competing interests:** The authors have declared that no competing interests exist.

the use of telerehabilitation in service delivery. Costs of intervention delivery and training requirements were poorly reported.

## Conclusions

This review synthesises the evidence relating to factors that may influence stroke telerehabilitation intervention delivery at a crucial timepoint given the rapid deployment of telerehabilitation in response to the COVID-19 pandemic. It recommends strategies, such as ensuring adequate training and technical infrastructure, shared learning and consistent reporting of cost and usability and acceptability outcomes, to overcome challenges in embedding and routinising this service model and priorities for research in this area.

## Introduction

Rehabilitation is one of the most important aspects of care following a stroke, leading to better recovery and higher levels of independence [1]. Globally the prevalence of stroke has increased by 85% in the last thirty years, and it now represents the condition with the highest need for rehabilitation worldwide [2]. This means that increasingly, despite positive evidence for post stroke rehabilitation [1, 3], the recommended amount of therapy is rarely available or achieved [4] resulting in unmet ongoing rehabilitation needs [2, 5]. This limitation on therapy is further compounded by the COVID-19 pandemic, causing widespread disruption of healthcare services and concern that healthcare facilities could be sources of contagion [6, 7]. Given the disease transmission mechanism and requirement to reduce in-person contacts, including between patient and clinician, telerehabilitation offers a unique solution allowing convenient access to post-stroke rehabilitation without exposure risk [6, 7]. It has been recommended to prevent service interruption, where quarantine or social distancing measures have been advised [8, 9]. In addition to its current necessity in response to COVID-19, telehealth may continue to contribute to the solution to longstanding limitations on therapy. It may free up clinician time and address some barriers faced by people with stroke such as time restraints, geographical isolation and compliance [10].

Telerehabilitation is a branch of telehealth including the provision of rehabilitation services to patients at a remote location using information and communication technologies across distance or time [11, 12]. Several recent reviews supporting telerehabilitation for stroke rehabilitation compared to in-person care, are centred around clinical effectiveness [10, 13–15]. Despite the available evidence regarding the effectiveness of telerehabilitation, there has been little focus on factors influencing telerehabilitation delivery or its translation from the research setting into stroke practice, including technical requirements, challenges, practicalities, and factors related to usability and acceptability. The latter two are known factors that impact on digital intervention uptake and continued use [16]. Given the varied degree of impairments and activity limitations experienced post-stroke, for example impacting motor function, cognitive function and communication [3], additional considerations may be required for telerehabilitation post-stroke to ensure accessibility and engagement. Given the opaque timeline of COVID-19 and future longer-term disruptions to stroke rehabilitation services, it is crucial that these factors are explored. Given the well-established evidence, the intention of this systematic review is not to provide definitive conclusions regarding the effectiveness of stroke telerehabilitation. The intention is to search and synthesise the evidence regarding the practicalities of delivering telerehabilitation in stroke care. This will help design appropriate

interventions and identify factors to be considered to enable stroke telerehabilitation reach its full potential.

## Aims and objectives

The aim was to identify and describe the scientific literature relating to factors influencing the delivery of stroke telerehabilitation interventions. The specific objectives included:

1. To synthesise intervention delivery, including platforms used, dose and technical requirements.

2. To summarise training and support requirements for intervention delivery.

3. To explore factors relating to access of telerehabilitation in this clinical area and the cost of delivering telerehabilitation.

4. To explore the usability and acceptability of stroke telerehabilitation interventions, including participant-reported outcomes, adherence, adverse events, and facilitators and barriers to use.

## Methods

The protocol was developed a priori according to the Preferred Reporting Items for Systematic Reviews and Meta-Analyses (PRISMA) guidelines and registered on Prospero (CRD42020186024). The registered protocol uses the term rapid review; however, on reflection, this is a systematic review given the comprehensive data extraction and synthesis.

### Data sources and searches

Studies were initially identified from a recent Cochrane review on telerehabilitation services for stroke [13] which identified papers up to December 2018. This was supplemented by our search of five electronic bibliographic databases (MEDLINE, EMBASE, CINAHL, Web of Science and Cochrane Library and Central Registry of Clinical Trials) between January 2019 to May 2020, to identify papers published since the Cochrane review [13].

Predefined search strategies, based on those used in the Cochrane review [13], were developed with assistance of a librarian and piloted prior to use. The Medline search strategy can be found in S1 File.

Reference lists of eligible studies were hand-searched, citation tracking of these publications conducted and a Google Scholar search performed to identify additional studies missed in the original searches.

The search results were imported into ProQuest RefWorks bibliographic software and duplicate studies removed. Screening was divided amongst the reviewers using the Covidence systematic review software. Titles and abstracts and full text papers of potentially relevant studies were screened by two independent reviewers. Conflicts were decided by an independent verifier.

### Study selection

This review included full-text articles of randomised controlled trials (RCTs) and protocols for RCTs published in English, that delivered telerehabilitation interventions to people with stroke.

This review included adult stroke survivors with all types of stroke, at all levels of severity, and at all stages post-stroke (acute, subacute, or chronic). It excluded studies involving a mixture of stroke and non-stroke participants where data about stroke participants was not reported separately. Trials with children were excluded given the low stroke incidence and the additional challenges to delivering therapy via technological means in this paediatric population.

For the purposes of this review, telerehabilitation was defined as the provision of rehabilitation services, including assessment, review or rehabilitation, to patients at a remote location using information and communication technologies [11]. Interventions where telerehabilitation was not a major component were excluded, judged by team consensus, e.g. intervention included only one telerehabilitation session; the participants received more in-person than telerehabilitation contact; the only telerehabilitation component was either automated, not monitored by clinician/researcher, or only a helpline if required.

There were no restrictions related to clinical outcomes or context, as we were interested in interventions performed in all settings and geographical locations, and delivered by all types of therapists or non-therapists.

## Data extraction and quality assessment

Study details and data were extracted using a customised form, developed based on an eHealth checklist [17], and piloted prior to use (S2 File). The form was used to capture information related to demographics, intervention/control arm details, outcomes and results. Telerehabilitation intervention delivery, such as platforms used, dose and technical requirements were extracted. Human support and training related to the delivery of telerehabilitation required for participants, their carers and clinicians delivering the telerehabilitation was extracted. We extracted data related to access, such as relevant eligibility criteria and requirements for inclusion in the studies, and costs. Usability and acceptability data extracted consisted of participant-reported outcomes, including data related to usability, acceptability and satisfaction from the patient, carer or clinician perspective; adherence-related outcomes, such as usage of systems, completion of sessions and engagement with rehabilitation; safety and adverse events; and facilitators and barriers to use. Data extraction was completed independently for each paper by one of two reviewers (50% AS, 50%SH), with 20% checked by another reviewer (10% KP, 10% SM).

Given the range of telehealth approaches being used across the studies, the research team agreed the following definitions: *synchronous telerehabilitation* was used to describe interventions with real-time clinician-patient interaction during real-time review of the rehabilitation activity; *asynchronous telerehabilitation* was used to describe interventions where rehabilitation activity was conducted independently by the patient and their progress reviewed later by the therapist with a follow-up clinician-patient interaction to review rehabilitation progress; and, *telesupport* was used to describe interventions that provided patients only with support, advice or education related to their stroke. Technical support or helplines were not categorised as *telesupport*, for these purposes. Where interventions delivered more than one type of telerehabilitation, they were categorised based on the greatest component of the intervention. Definitions and categorisation according to definitions were agreed by consensus within the research team.

Risk of bias was assessed (for completed studies only) by a single reviewer, verified by a second. Where agreement could not be achieved with discussion, a third reviewer completed a consensus assessment. We assessed each study using the Cochrane Risk of Bias tool (V1) [18] for consistency with Laver [13], grading on each criterion as having low, high, or unclear risk

of bias. A study was judged to be at low risk of bias overall when all domains had a low risk of bias. Conversely, a study was judged to have a high risk of bias when it reported a feature judged as high risk of bias in any domain.

### Data synthesis and analysis

A narrative synthesis was conducted, providing a comprehensive description of the telerehabilitation interventions for stroke rehabilitation. The synthesis explored factors relating to delivery of the interventions, such as platforms used and technical requirements, training and support required, access and costs and other facilitators and challenges to implementation. The synthesis described patterns in adherence, usability and acceptability and explored factors that may contribute to any differences across the included studies.

### Role of the funding source

The funders played no role in the design, conduct, or reporting of this study.

## Results

The PRISMA flow diagram (Fig 1) summarises the study selection process. The search strategy identified 2092 results. An additional 266 records were identified by hand searching. Of these, the full texts of 159 studies were screened for inclusion. This review included 31 studies [19–49] and ten protocols of ongoing studies [50–59], adding eleven studies and five protocols to the recent Cochrane review [13].

See Table 1 for study characteristics of included RCTs, which included a total of 3368 participants, ranging from 10 to 573 participants, with 58% male. The majority of studies were conducted in the USA (n = 8).

### Telerehabilitation interventions

**Intervention delivery.** Comprehensive descriptions of the interventions can be found in S3 File. Table 2 provides a summary of telerehabilitation intervention characteristics. The rehabilitation aims of the interventions and type of telerehabilitation delivered varied. Most interventions were aimed at improving a stroke primary or secondary impairment: physical function (n = 10); upper limb function (n = 5); speech and language ability (n = 3); cognitive function (n = 2); and visual impairment (n = 1). Others targeted self-management, including secondary prevention and health behaviour change (n = 4), quality of life (n = 4) and mental health (n = 2).

The interventions that aimed to improve a stroke primary or secondary impairment (n = 21) were delivered by *asynchronous telerehabilitation*, where the participant completed self-led therapy which was reviewed by remote consultation with the clinician (n = 10), real-time *synchronous telerehabilitation*, where the clinician remotely supervised the participants' therapy in real-time (n = 9), or *telesupport*, where remote consultations were for education, support, or goal-setting only (n = 2).

One intervention for self-management delivered *asynchronous telerehabilitation* via a mobile app and online chat [28]. The remaining interventions aimed at self-management, quality of life and mental health, were delivered by *telesupport* only.

**Platforms used.** Most studies used either telephone (n = 15) or video (n = 15) call as a mode of communication within the telerehabilitation interventions; one of which compared videoconference versus telephone delivery [32]. Nine interventions involved telephone only [20, 21, 30, 31, 36, 42, 43, 47]; these tended to be interventions delivering *telesupport*.

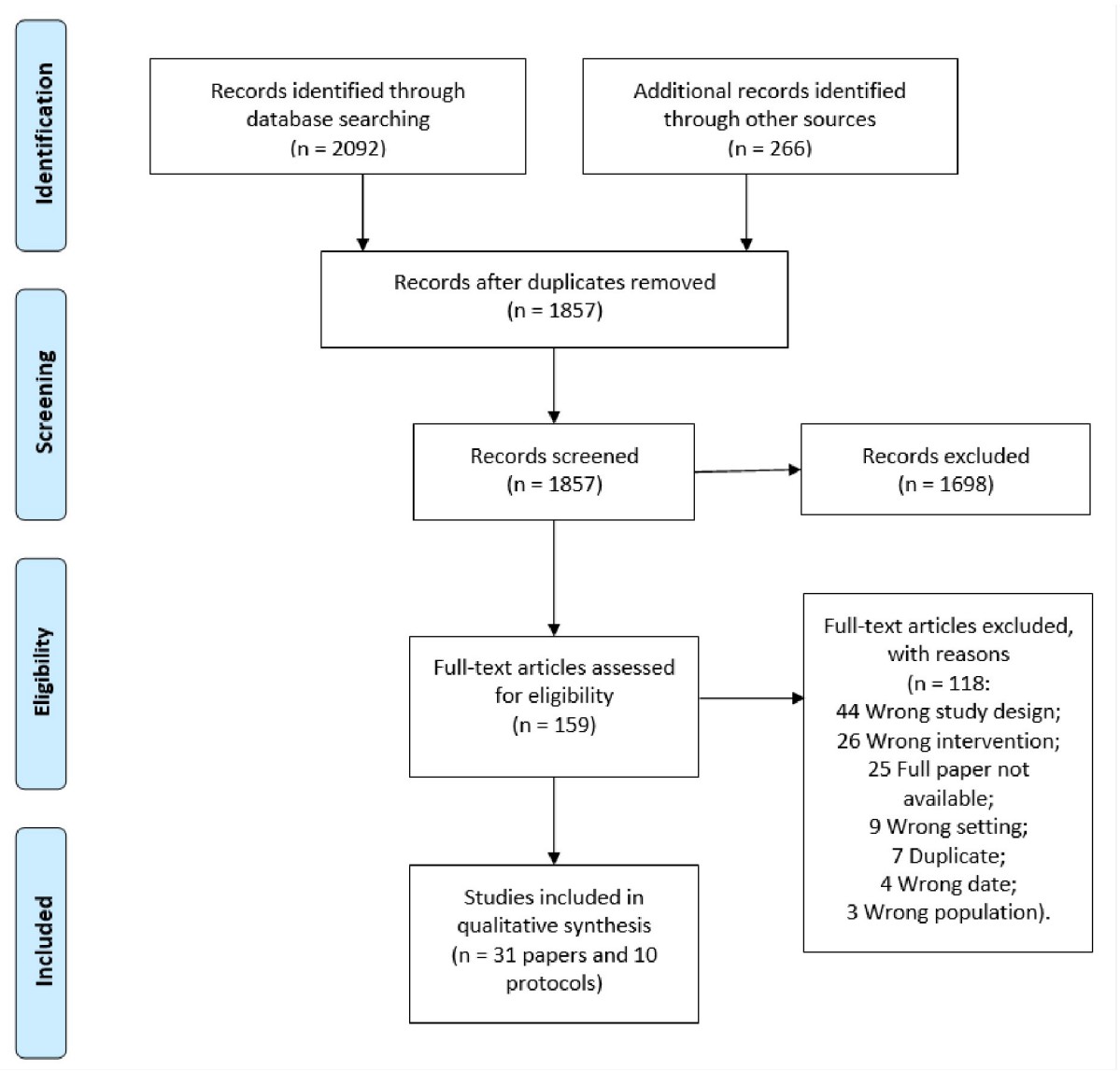

**Fig 1. PRISMA 2009 flow diagram.**

Telesupport was also delivered via online communication, such as email and online forum [44] or text reminder system plus telephone call [48].

The remaining n = 22 interventions included a combination of telerehabilitation components. Videoconference along with a digital component, such as a computer-, tablet- or app-based component, was used in n = 11 studies [19, 22, 23, 25, 26, 29, 33, 35, 37, 39, 40]. Telephone along with a digital component was used in n = 5 studies [24, 34, 45, 48, 49].

**Dose.** Intervention duration was similar in interventions delivered by *synchronous telerehabilitation* (range = 4–12 weeks), and a*synchronous telerehabilitation* (range = 10 days to 12 weeks). However, the frequency of contact tended to be higher with *synchronous telerehabilitation* (once/week to twice/day), compared with a*synchronous telerehabilitation* (once/week to twice/3 months). *Telesupport* interventions tended to be of longer duration; almost half were

**Table 1. Table of study characteristics.**

| Study ID Country | Participants N M/F Mean age Time post-stroke | Intervention Including telerehabilitation component and digital and non-digital co-interventions | Control | Outcomes of interest Primary clinical outcome and outcomes relevant to usability (adherence, satisfaction) | Key findings Primary clinical outcome and outcomes relevant to usability (adherence, satisfaction) |
|---|---|---|---|---|---|
| Asano 2019 Singapore | Total n = 124 (IG n = 61; CG n = 64) 65M/59F 64.1 years <4 weeks post-stroke | Rehabilitation exercises via tablet-based telerehabilitation system plus video-conferenced reviews. | Usual rehabilitation care. | Disability component of the Late-Life Function and Disability Instrument Participation in rehabilitation and exercise Other usability outcome NR. | NSD in improvements in the functional outcomes between the IG and CG at three months post intervention. NSD in the time spent on rehabilitation and exercise between the two groups. |
| Bishop 2014 USA | Total n = 49 stroke survivor-carer dyads (IG n = 23, CG n = 26) 17M/32F 70.1 ± 11.6 years <6 months post-stroke | Telephone consultation with survivors and carers separately to identify and address problems, provide education, facilitate problem solving, and provide follow-up support. Each dyad was provided written information and resources. | Usual medical follow up. | Primary analysis was focused on 3 global outcome scores: health care utilisation, family functioning, and general functioning. Adherence and usability outcomes NR. | IG significantly decreased overall health care utilisation, improved family functioning and general functioning, and improved stroke survivor and carer quality of life. |
| Boter 2004 Netherlands | Total n = 536 (IG n = 263; CG n = 273) 260M/276F Median (IQR) age in IG 66 (52–76) and CG 63 (51–74) Time post-stroke NR | Outreach care program on stroke prevention, stroke services and individualised support via 3 telephone calls and 1 home visit. | Usual care. | SF-36 Dissatisfaction with care Adherence NR. | IG had better scores on the SF-36 domain "Role Emotional" than CG. In both groups, one-fifth of the patients were dissatisfied with care received in the hospital, and half were dissatisfied with care received after discharge, with NSD between IG and CG. |
| Carey 2007 USA | Total n = 20 (IG1 n = 10, IG2 n = 10) 15M/5F 66.7 ± 9.6 years >12 months post-stroke | Both groups received TR via a laptop using customised software and custom-made electro-goniometer braces and potentiometers with the aim of practicing finger and wrist movements. Regular teleconferencing (approx. 5 sessions in two weeks) between therapist and participant. IG1: tracking software provided feedback and an accuracy score. IG2: tracking software showed a sweeping cursor representing movement, but no other feedback. IG2 crossed over to receive an additional 2 weeks of IG1 tracking training. | See "Intervention" column. | Behavioural changes were measured with the Box and Block test, Jebsen Taylor test, and finger range of motion, and finger-tracking activation paradigm during functional MRI. Adherence and usability outcomes NR. | IG showed significant improvement in all 4 behavioural tests; CG improved in the Box and Block and Jebsen Taylor tests. NSD between groups in improvement in the Box and Block and Jebsen Taylor tests. CG, after crossing over, did not show further significant improvements. |
| Chen 2017 China | Total n = 54 (IG n = 27; CG n = 27) 33M/21F 66 years 14–90 days post-stroke | Telerehabilitation system with exercise and electromyography-triggered neuromuscular stimulation supervised by videoconference. | Same therapeutic strategy delivered in-person in conventional outpatient rehabilitation setting. | Modified Barthel Index (MBI) to measure disability and activities of daily living Adherence and usability outcomes NR. | Both groups showed significant improvements after treatment, with no difference in the groups at any time point. |

(*Continued*)

**Table 1.** (Continued)

| Study ID Country | Participants N M/F Mean age Time post-stroke | Intervention Including telerehabilitation component and digital and non-digital co-interventions | Control | Outcomes of interest Primary clinical outcome and outcomes relevant to usability (adherence, satisfaction) | Key findings Primary clinical outcome and outcomes relevant to usability (adherence, satisfaction) |
|---|---|---|---|---|---|
| Chumbler 2012 USA | Total n = 48 (IG n = 25; CG n = 23) 47M/1F 67.4 ± 9.6 years <2 years post-stroke | Televisits where researcher video recorded the home environment and the participant completing tests of physical and functional performance that were later reviewed by the teletherapist, in-home messaging device, and telephone call reviews plus routine care as directed by their providers. | Usual care. | Motor subscale of the Telephone Version of Functional Independence Measure, function scales of the Late-Life Function and Disability Instrument Adherence and usability outcomes NR. | IG improved at 6 months and CG declined, but the differences were NSD. |
| Cramer 2019 USA | Total n = 124 (IG n = 62, CG n = 62) 61 ± 13.49 years 90M/34F 4–36 weeks post-stroke | Intensive arm motor therapy via an in-home internet-connected computer, including exercises, functional training (including games) and stroke education guided by the TR system. Half of the sessions included videoconference with the therapist via the TR system. Participants signed a behavioural contract including a treatment goal, and treatment was based on an upper extremity task-specific training manual and accelerated skill acquisition program. | Same intensity, duration, and frequency of therapy and stroke education content but provided in clinic with therapist feedback based on observations on supervised days. | Fugl-Meyer Upper Extremity Scale Adherence Patient Satisfaction Questionnaire Physical Activity Enjoyment Scale Optimisation in Primary and Secondary Control scale | IG experienced substantial gains in arm function; not inferior to CG. 98.3% adherence in IG; NSD in adherence between IG and CG. Both groups reported high satisfaction; slightly higher in CG than IG (mean [SD], 55.2 [7.7]vs 58.5 [8.0]; P = .02). NSD between groups post-intervention. Higher motivation (0.47 points) in the CG (P = 0.008) post-intervention. |
| Deng 2012 USA | Total n = 16 (IG1 n = 8; IG2 n = 8) 54.7 ± 12.5 years 11M/5F >5 months post-stroke | Both groups received same dose of TR to practice ankle movements via a laptop using customised tracking software without direct supervision by the therapist, with remote monitoring and teleconferencing. IG1: tracking software provided feedback and an accuracy score. IG2: tracking software showed a sweeping cursor representing movement, but did not provide the target or response or an accuracy score. IG2 crossed over to receive an additional 2 weeks of IG1 tracking training. | See "intervention" column. | Paretic ankle dorsiflexion during the swing phase of gait measured using surface markers and an 8-camera motion capture system (Vicon). Adherence Qualitative feedback collected | Dorsiflexion during gait was significantly larger in IG1 compared with IG2. 16/19 participants completed all the training. Favourable for TR. More detail in "intervention characteristics" table. |
| Forducey 2012 USA | Total n = 11 (2 lost post-randomisation, IG n = 4; CG n = 5) 60 years (range 47–75) 6M/5F <6 months post-stroke | Desktop videophone communication with therapist (OT and PT) for education, retraining of self-care, functional mobility and posture, home modifications and therapy to improve function in impaired limbs, plus provided with written material on stroke risk factors, warning signs, and community-based support groups. | Same content delivered by in-person home health care (PT and OT). | Functional Independence Measure, SF-12 Adherence and usability outcomes NR. | Significant pre-post differences were found for both the IG and CG on the FIM and SF-12. The IG required significantly fewer visits to achieve clinically meaningful outcomes. |

*(Continued)*

**Table 1.** (Continued)

| Study ID Country | Participants<br>N<br>M/F<br>Mean age<br>Time post-stroke | Intervention<br>Including telerehabilitation component and digital and non-digital co-interventions | Control | Outcomes of interest<br>Primary clinical outcome and outcomes relevant to usability (adherence, satisfaction) | Key findings<br>Primary clinical outcome and outcomes relevant to usability (adherence, satisfaction) |
|---|---|---|---|---|---|
| Grau-Pellicer 2020<br>Spain | Total n = 41 (IG n = 24; CG n = 17)<br>65.27 ± 11.91 years<br>24M/17F<br>Mean time post-stroke >18 months | Digital platform with mHealth apps to supervise adherence to physical activity, group rehabilitation program, ambulation program at home, and WhatsApp group. | Conventional rehabilitation including trunk exercises, muscle strengthening, occupational therapy and gait training. | Physical activity measured using Community Ambulation and Sedentary Behaviour<br>Adherence<br>Satisfaction via ad hoc questionnaire.<br>Other usability | Community ambulation and sitting time improved more in IG than in the CG.<br>Participant-reported use of the app: 50% of participants were able to use the app.<br>100% would recommend the program; 91% "very satisfied" with the program.<br>68.2% preferred combination of app and exercise program, compared with 27% preferring exercise program and 4.5% preferring the app. |
| Huijgen 2008<br>Netherlands | Total n = 16 (IG n = 11; CG n = 5)<br>50 ± 18<br>39M/16F<br>Mean time post-stroke >18 months | TR for arm/hand function using the Home Care Activity Desk training at home. Therapy was video recorded; videos and the results of the exercises were uploaded to the hospital server. Reviewed remotely by therapist for weekly videoconference with the patient. | Usual care and generic exercises prescribed by the physician. | Action Research Arm Test, Nine Hole Peg Test<br>Adherence<br>User satisfaction visual analogue scale completed by participants and therapists | Both IG and CG maintained or improved their arm/hand function; NSD between the IG and CG.<br>Mean treatment time:<br>IG = 9.5hours per month;<br>CG = 9hours per month.<br>Both participants and therapists were satisfied with the system. |
| Joubert 2020<br>Australia | Total n = 249 (IG n = 112; CG n = 137)<br>Median (range) 72 (36–103) years<br>"Roughly equal" M/F. Details NR.<br><3 months post-stroke | Integrated care model including telephone follow-up between care-coordinator and stroke survivor, carer and family. Also included in-hospital education and written information regarding stroke mechanism, stroke risk factors, and follow-up procedure and a 3-monthly planned review by primary care physician. Primary care physician received ongoing support and advice from the stroke specialist. Exchange of data (risk factors, depression, social factors) between physician and the Stroke Service. | Usual care by primary care physicians. | Improvement or abolition of risk factors such as raised blood pressure, diabetes, hyper-lipidaemia, the modification of adverse life-style factors such as lack of exercise, smoking and alcohol abuse and adherence to preventive medication at one year.<br>Adherence and usability outcomes NR. | IG experienced greater improvement than CG in risk factors, such as hypertension, alcohol abuse, smoking, BMI and exercise tolerance. |
| Kirkness 2017<br>USA | Total: n = 100 (IG n = 37, CG1 n = 35, CG2 n = 28)<br>50M/50F<br>60.3 (range 23–88) years<br><4 months post-stroke | Brief psychosocial behavioural intervention delivered via one in-person orientation session followed by six telephone sessions with psychosocial nurse practitioner therapist plus usual primary care or stroke provider stroke follow-up care. | CG1: Same intervention delivered in-person (usually in the participant's home). CG2: Usual care. | Hamilton Depression Rating Scale (HDRS)–response (% reduction) and remission (score <10).<br>Adherence and usability outcomes NR. | A brief psychosocial intervention delivered by telephone (IG) or in-person (CG1) did not reduce depression significantly more than usual care (CG2). |

(*Continued*)

**Table 1.** (Continued)

| Study ID Country | Participants N M/F Mean age Time post-stroke | Intervention Including telerehabilitation component and digital and non-digital co-interventions | Control | Outcomes of interest Primary clinical outcome and outcomes relevant to usability (adherence, satisfaction) | Key findings Primary clinical outcome and outcomes relevant to usability (adherence, satisfaction) |
|---|---|---|---|---|---|
| Li 2020 China | Total n = 120 (IG1 n = 60; IG2 n = 60) 49M/71F 59.7 years Mean time post-stroke 90 days | IG1: Post-discharge assessment of functional tasks via videoconference. IG2: Post-discharge assessment of functional tasks via telephone. | See "intervention" column. | Validity and reliability of functional assessments delivered via videoconference versus telephone (compared with gold-standard home visit). Completion rates of remote assessments Acceptability (satisfaction, comfort, confidence were rated on a 4-point scale) | Videoconference, but not telephone, administration was as valid and reliable as in-person, home visit assessment at both 2-week and 3-month follow-up periods. Completion rates of both IG1 and IG2 were >80% at all follow-up periods. IG1 reported higher satisfaction and confidence using the videoconference assessment to measure their functional status than IG2 telephone assessment. |
| Lin 2014 Taiwan | Total n = 24 (IG N = 12; CG n = 12) 17M/7F 75.1 ± 2.9 years >6 months post-stroke | Balance training delivered remotely via videoconference with therapist. | Conventional balance training program delivered in-person with 2 patients to 1 therapist. | Berg Balance Scale Adherence NR. Satisfaction | Both the IG and CG had significant improvement on BBS score; however, NSD was observed between the two groups. Good level of satisfaction in both groups; not comparable due to different measures used. |
| Llorens 2015 Spain | Total n = 30 (IG N = 15; CG n = 15) 17M/13F 55.5 ± 8.4 years >6 months post-stroke | Kinect balance training at home with telephone review + conventional in-clinic physiotherapy not related to balance. | Kinect balance training in-clinic + conventional in-clinic physiotherapy not related to balance. | Berg Balance Scale Adherence NR. System Usability Scale (SUS) Intrinsic Motivation Inventory (IMI) | Both the IG and CG had significant improvement on BBS score; however, NSD was observed between the two groups. SUS mean scores in both groups were high: IG 87.5±5.4 and CG 85.4±4.7, with NSD between groups. IMI positive scores in all domains: interest/ enjoyment, perceived competence, pressure/ tension, and value/ usefulness, with NSD between groups. |
| Maresca 2019 Italy | Total n = 30 IG n = 15, CG n = 15 51.2 ± 11.3 years 14M/16F Time post-stroke NR | Phase 1 (during hospital admission) tablet-based virtual reality rehabilitation system (VRRS) with cognitive and speech modules for aphasia rehabilitation. Phase 2 at home use of VRRS-Tablet with videoconference review. | Phase 1 (during hospital admission) traditional linguistic treatment. Phase 2 usual care including conventional speech therapy. | Neuropsychological evaluation including Token Test (language), Esame Neurologico Per l'Afasia (language), Aphasic Depression Rating Scale, EQ-5D and Psychosocial Impact of Assistive Devices Scale. Adherence and usability outcomes NR. | IG improved in all the investigated areas, except for writing, while the CG only improved in comprehension, depression, and quality of life. |
| Mayo 2008 Canada | Total n = 190 (IG n = 96; CG n = 94) 116M/74F 70.99 ± 13.76 years Time post-stroke NR | Case management intervention–telephone intervention for post-discharge management including communication with participants' physicians. | Usual care—Participant and family advised to contact their physician. | HRQoL using the Physical Component Summary of the SF-36. Adherence and usability outcomes NR. | Both the IG and CG had significant improvement in HRQoL; however, NSD was observed between the two groups at any timepoint. |

(*Continued*)

**Table 1.** (Continued)

| Study ID Country | Participants N M/F Mean age Time post-stroke | Intervention Including telerehabilitation component and digital and non-digital co-interventions | Control | Outcomes of interest Primary clinical outcome and outcomes relevant to usability (adherence, satisfaction) | Key findings Primary clinical outcome and outcomes relevant to usability (adherence, satisfaction) |
|---|---|---|---|---|---|
| Meltzer 2018 Canada | Total n = 55 recruited, n = 44 analysed (IG1 n = 17, IG2 n = 6, CG1 n = 16, CG2 n = 5) 27M/17F 64.2 ± 10.8 >6 months post-stroke | Computerised speech and language exercises + remote therapy sessions via teleconference. | Computerised speech and language exercises + in-person therapy sessions. | Western Aphasia Battery aphasia quotient (objective language impairment). Adherence and usability outcomes NR. | Both the IG and CG had significant improvement; however, NSD was observed between the groups. This demonstrated non-inferiority of telerehabilitation. |
| Ora 2020 Norway | Total n = 62 (IG n = 32 CG n = 30) 41M/21F 64.85 ± 11.85 Included any time post-stroke | Augmented language training via videoconference. | Usual care. | Norwegian Basic Aphasia Assessment: naming Adherence and usability outcomes NR. | NSD was observed between the groups at four weeks and four months post-randomisation. |
| Piron 2008 Italy | Total n = 10 (IG n = 5; CG n = 5) 5M/5F IG: 53 ± 15 years CG: 65 ± 11 years Mean time post-stroke 13 months | Virtual reality upper limb rehabilitation program with videoconferencing observation. | Virtual reality upper limb rehabilitation program in presence of a physiotherapist. | Participant satisfaction questionnaire. Secondary outcome: Fugl-Meyer Upper Extremity Scale. Adherence NR. | IG satisfaction scores were equal to or higher than the CG in all items investigated. IG and CG improved significantly in Fugl-Meyer score. |
| Piron 2009 Italy | Total n = 36 (IG n = 18; CG n = 18) 21M/5F 65.2 ± 7.8 years 7–32 months post-stroke | Virtual reality motor tasks via 3D motion tracking system with videoconferencing observation and feedback by clinician | Conventional upper limb physiotherapy in-person. | Fugl-Meyer Upper Extremity Scale Adherence and usability outcomes NR. | Significant improvements in both groups, maintained at follow-up. Moderate effect of telerehabilitation compared with conventional therapy. |
| Rochette 2013 Canada | Total n = 186 (IG n = 92; CG n = 94) 107M/79F 62.5 ± 12.5 years <1 month post-stroke | Multimodal (telephone, Internet, and paper) support intervention (participants contacted by clinician). | Participants provided with contact details of a trained healthcare professional to contact if required. | Unplanned use of health services for an adverse event, Quality of Life Index and EQ-5D. Adherence and usability outcomes NR. | No significant differences between the IG and CG on unplanned use of health services and quality of life. The quality of life measures improved significantly for both groups. |
| Rodgers 2019 UK | Total n = 573 (IG n = 285; CG n = 288) 342M/131F 71 years Median time post-stroke 73 days | Extended stroke rehabilitation service via telephone reviews. | Usual care. | Nottingham Extended Activities of Daily Living Scale (NEADL) Adherence Experience of services survey Cost | NSD in improvements in NEADL scores between the IG and CG. 86% of expected reviews were completed. IG appeared to be more satisfied with some aspects of care. Intervention was associated with cost saving. |
| Saal 2015 Germany | Total n = 265 (IG n = 130; CG n = 135) 137M/128F 68.25 ± 12.63 years Time post-stroke NR | Stroke support service focused on counselling and referral, including home visit and telephone calls, educational sessions (in-person) and written patient information on disease-specific and care-related issues plus usual care (physician, specialist care, rehabilitation therapy and care). | Usual care plus two brochures containing general information on risk factors and warning signs for strokes. | Stroke Impact Scale version 2.0, physical function sub-scale Adherence and usability outcomes NR. | The intervention did not positively influence physical function or the secondary endpoints (depression, recurrence of stroke or HRQoL). The data suggested a reduced risk of mortality in the intervention group; however, this was not a primary outcome. |

(*Continued*)

**Table 1.** (Continued)

| Study ID Country | Participants N M/F Mean age Time post-stroke | Intervention Including telerehabilitation component and digital and non-digital co-interventions | Control | Outcomes of interest Primary clinical outcome and outcomes relevant to usability (adherence, satisfaction) | Key findings Primary clinical outcome and outcomes relevant to usability (adherence, satisfaction) |
|---|---|---|---|---|---|
| Smith 2012 USA | Total n = 38 carer-stroke survivor dyads 38M stroke survivors /38F carers Mean age NR Time post-stroke NR | Web-based conferencing and video education intervention with online library resource. | Access to online library resource only. | Center for Epidemiologic Studies Depression Scale (CES-D) Perceived Credibility, Effort, and Benefit - Credibility/Expectancy Questionnaire Adherence NR. | No statistically significant effect on depression among SSs. Carers in the IG reported significantly lower depressions than those in the Control. Carers in the IG reported higher usefulness, likelihood to recommend and greater benefit to SS than carers in the CG. NSD between groups in perceived effort or benefit to carers. |
| Svaerke 2019 Denmark | Total n = 18 (IG n = 9; CG n = 9) 7M/7F completed 64.4 ± 11.6 years 3–42 days post-stroke | Computer-based cognitive rehabilitation–early intervention plus weekly telephone review. | Usual care then computer-based cognitive rehabilitation–late intervention. | Neuropsychological test battery–tested lateralised visuospatial symptoms using Cognitive Assessment at Bedside with iPad. Adherence and usability outcomes NR. | Intervention improved visuospatial symptoms after stroke significantly when administered early in the sub-acute phase after stroke. Improvement not maintained 3 weeks post-intervention. |
| Torrisi 2019 Italy | Total n = 40 (IG n = 20; CG n = 20) 26M/14F 55.2 ± 18.4 years 3–6 months post-stroke | Phase 1 (during hospital admission) cognitive rehabilitation training using tablet-based virtual reality rehabilitation system (VRRS-Evo). Phase 2 at home use of VRRS Home Tablet with videoconference review of progress by psychologist. | Phase 1 (during hospital admission) same exercises using paper–pencil tools. Phase 2: the control group continued the traditional training, with the same amount of treatment. | Montreal Cognitive Assessment, attentive matrices, Trail Making Test B, Phonemic Fluency, Semantic Fluency, Rey Auditory Verbal Learning Test I, Hamilton Rating Scale-Anxiety and Hamilton Rating Scale-Depression. Adherence and usability outcomes NR. | Significant improvement in the global cognitive level, as well as in the attentive, memory and linguistic skills in the IG. On the other hand, NSD were found in executive function. IG showed a significant decrease in anxiety compared to CG. |
| Wan 2016 China | Total n = 91 (IG n = 45; CG n = 46) n = 80 reported in results. 57M/23F 59.66 ± 12.40 years <1 month post-stroke | Goal-setting telephone follow-up program for self-management and health behaviour change plus usual stroke education and care. | Usual stroke education and care only. | Modified health behaviour scale (25 items; 6 sub-categories: PA, nutrition, low-salt, smoking, alcohol, BP, medication adherence), developed from 2 subscales of the Health Promoting Lifestyle Profile II (HPLP II) and 4 stroke-related subcategories Adherence and usability outcomes NR. | Both groups' health behaviour improved, but NSD between groups at any timepoint except for improved medication adherence in the intervention group at 6 months. |
| Wang 2020 China | Total n = 174 (IG n = 87; CG n = 87) 100M/74F Age not reported Included any time post-stroke | Comprehensive reminder to improve patients' health behaviours–text messages plus monthly telephone follow-up interviews by nurses. | Usual care plus stroke prevention handout plus 2 x telephone call with nurse within first month. | Health Promoting Life-style Profile II—used to assess the health behaviours Adherence and usability outcomes NR. | The intervention improved health behaviours and medication adherence and reduced blood pressure and disability; maintained 6 months post-discharge. |
| Withiel 2019 Australia | Total n = 65 (IG n = 22; CG1 n = 24; CG2 n = 19) 38M/27F 60.9 ± 12.8 years >3 months post-stroke | Computerised cognitive training using Lumosity. Weekly compliance monitored remotely. Weekly telephone contact for compliance. | CG1—Memory skills group training CG2 –Waitlist control | Goal Attainment Scale for memory specific rehabilitation goals Adherence Usability outcomes NR. | Memory group training (CG1) were more likely to achieve their memory improvement goals than IG. Adherence rate was 83% in both IG and CG1. |

IG–intervention group; CG–control group; F–female; M—male; NSD–no significant difference; NR–not reported; HRQoL–health-related quality of life; EQ-5D – EuroQOL-5D; SF-36—Short Form 36-item; SF-12 –Short Form 12-item.

**Table 2. Summary of telerehabilitation intervention characteristics.**

| Study ID | ST/ AT/ TS | Telerehabilitation intervention components | Dose | | | | Reported | Setting | Training | | Additional support | | |
|---|---|---|---|---|---|---|---|---|---|---|---|---|---|
| | | | Intervention duration | Number of sessions | Session duration | Other | | | Dose | Training description | Clinician | Carer | Technical |
| **Physical function (n = 10)–functional independence (n = 6), balance (n = 2), gait (n = 1), assessment (n = 1)** | | | | | | | | | | | | | |
| Asano 2019 | AT | Tablet-based rehab (remote monitoring); Videoconference; Physiological sensors | 3 months | 1/week video | NR | Therapy 5/ week | ✓ | During hospital admission | 1–3 x 1-hour sessions | Training; Competency check; Guide; videos | - | ✓ | - |
| Chen 2017 | ST | Videoconference; Physiological sensors | 12 weeks | 2/day | 80 min | NR | ✓ | During hospital admission | NR | Training plus practice until competent | ✓ | ✓ | - |
| Chumbler 2012 | AT | Video-recorded televisits; In-home messaging device; Telephone | 3 months | 3 tele-visits, 5 calls | NR | Daily IHMD use | NR | - | - | - | ✓ | - | - |
| Deng 2012 | AT | Videoconference; Computerised rehab (remote monitoring); Physiological sensors | 1 month | 2/ week contact | NR | 20 days training | ✓ | Outpatient | NR | Instruction plus practice until competent | - | ✓ | - |
| Forducey 2012 | ST | Videoconference | 6 weeks | 2/ week sessions | NR | NR | NR | - | - | - | - | ✓ | - |
| Li 2020 | ST | Videoconference or telephone via WeChat app | 3 months | 2 calls | NR | Assessment only | ✓ | During hospital admission | NR | Trained on use | - | - | - |
| Lin 2014 | ST | Computerised VR/ gaming therapy; Videoconference; Physiological sensors | 4 weeks | 3/ week | 50 min | - | ✓ | Not clear | NR | Operational technique training | ✓ | ✓ | - |
| Llorens 2015 | AT | VR/gaming therapy; Telephone | 7 weeks | 3/ week | 45 min | - | NR | - | - | - | ✓ | - | ✓ |
| Rodgers 2019 | TS | Telephone | 18 months | 5 calls | 2 hrs | Included indirect time | NR | - | - | - | - | ✓ | - |
| Saal 2015 | TS | Telephone | 12 months | 12 calls | NR | NR | NR | - | - | - | - | - | - |
| **Upper limb rehabilitation (n = 5)** | | | | | | | | | | | | | |
| Carey 2007 | AT | Computerised rehab (not monitored); Videoconference | 10 days | 5 contacts | NR | 10 sessions (2–8 h / day) | ✓ | Out-patient | One | Supervised practice; Video | ✓ | - | - |
| Cramer 2019 | ST | Videoconference; VR/gaming | 6–8 weeks | 18 sessions | 70 min | 18 sessions un-supervised | ✓ | Out-patient | NR | Trained to use system | - | - | ✓ |
| Huijgen 2008 | AT | Video-recorded therapy; Videoconference | 1 month | 1 / week video | NR | Therapy 30 mins x 5/week | ✓ | Out-patient | 4 sessions | Training sessions | - | - | - |

*(Continued)*

**Table 2.** (Continued)

| Study ID | ST/AT/TS | Telerehabilitation intervention components | Dose | | | | Reported | Setting | Training | | Additional support | | |
|---|---|---|---|---|---|---|---|---|---|---|---|---|---|
| | | | Intervention duration | Number of sessions | Session duration | Other | | | Dose | Training description | Clinician | Carer | Technical |
| Piron 2008 | ST | VR/ gaming; Videoconference | 1 month | 1 / day | 1 hour | - | ✓ | Not clear | NR | Briefly trained | - | - | ✓ |
| Piron 2009 | ST | VR/ gaming; Videoconference | 1 month | 5 / week | 1 hour | - | ✓ | Not clear | NR | Training | ✓ | - | ✓ |
| **Self-management (n = 4)–secondary prevention (n = 2), physical activity (n = 1), health behaviours (n = 1)** | | | | | | | | | | | | | |
| Grau-Pellicer 2020 | AT | Digital platform—app; Group chat (WhatsApp) | 8 weeks | NR | NR | In-person group | ✓ | Out-patient | NR | Training and supervised use as required | ✓ | - | - |
| Joubert 2020 | TS | Telephone | 12 months | Dependent on risk | NR | - | NR | - | - | - | - | ✓ | - |
| Wan 2016 | TS | Telephone | 3 months | 3 calls | 15–20 min | NR | NR | - | - | - | - | - | - |
| Wang 2020 | TS | Telephone calls; Text reminder system | 6 months | 4 calls | NR | Weekly texts | NR | - | - | - | - | - | - |
| **Quality of life (n = 4)–QOL and health service use (n = 3), QOL and satisfaction with care (n = 1)** | | | | | | | | | | | | | |
| Bishop 2014 | TS | Telephone | 3.5 months | 13 calls | 15 min | NR | NR | - | - | - | - | - | - |
| Boter 2004 | TS | Telephone | 5 months | 3 calls | NR | 1 home visit | NR | - | - | - | - | ✓ | - |
| Mayo 2008 | TS | Telephone | 6 weeks | 7.8 calls | 5–20 mins | Based on needs | NR | - | - | - | ✓ | ✓ | - |
| Rochette 2013 | TS | Telephone | 6 months | 1/week, 2/ month, 1/ month | 14.1±9.5 minutes | 10.3±5.4 minutes indirect time | NR | - | - | - | - | - | - |
| **Speech and language (n = 3)** | | | | | | | | | | | | | |
| Maresca 2019 | AT | Tablet-based rehabilitation; Videoconference | 12 weeks | 2 / week calls | NR | Therapy 50 mins x 5/week | ✓ | Inpatient | NR | Training | - | - | - |
| Meltzer 2018 | ST | Computerised therapy (iPad or computer); Videoconference | 10 weeks | 1 / week | 1 hour | - | ✓ | Outpatient | 2 hours | Instruction | - | ✓ | - |
| Ora 2020 | ST | Videoconference | 4 weeks | 5 / week | 1 hour | - | ✓ | Not clear | 30–60 mins | Training | - | - | - |
| **Cognitive function (n = 2)–cognitive function (n = 1), memory (n = 1)** | | | | | | | | | | | | | |
| Torrisi 2019 | AT | Tablet-based therapy (remote monitoring); Videoconference | 12 weeks | 2 / week video | NR | 3/ week therapy | NR | - | - | - | - | - | - |
| Withiel 2019 | AT | Telephone; Online computerised therapy (remote monitoring) | 6 weeks | 1 / week call | 30 mins | 5 / week training | NR | - | - | - | - | - | - |

(*Continued*)

**Table 2.** (Continued)

| Study ID | ST/ AT/ TS | Telerehabilitation intervention components | Dose | | | | Reported | Setting | Training | | Additional support | | |
|---|---|---|---|---|---|---|---|---|---|---|---|---|---|
| | | | Intervention duration | Number of sessions | Session duration | Other | | | Dose | Training description | Clinician | Carer | Technical |
| **Mental health–depression (n = 2)** | | | | | | | | | | | | | |
| Kirkness 2017 | TS | Telephone | 6 weeks | 6 calls | 10–80 mins | - | ✓ | At home or outpatient | 1 session | Orientation session / Manuals | - | ✓ | - |
| Smith 2012 | TS | Online communication (email, chat forum) | 11 weeks | 2 / week chats | NR | NR | ✓ | Self-led | NR | Online and hard copy tutorials | - | ✓ | ✓ |
| **Visual impairment (n = 1)** | | | | | | | | | | | | | |
| Svaerke 2019 | AT | Telephone / Online computerised therapy (not remotely monitored) | 3 weeks | 1 / week call | NR | Therapy every second day | ✓ | Not clear | NR | Instruction | - | - | - |

AT–asynchronous telerehabilitation; ST–synchronous telerehabilitation; TS–tele-support; NR–not reported; Min–minutes; H–hours; VR–virtual reality.

six months or longer [28, 30, 41–43], and patient-clinician interaction was less frequent in *tele-support* interventions (twice/week to five calls/18 months).

**Technical requirements.** Videoconferencing was primarily delivered via a laptop or personal computer with a webcam, using a variety of videoconferencing software: Skype [26]; WebEx or VSee [37]; Cisco/Jabber Acano [38]; or, not reported [22, 29, 33]. One study used the We Chat mobile app [32] and one used a desktop videophone [27]. The remaining eight studies used bespoke software accessed via a computer [23, 25, 39, 40] or a tablet [19, 35, 46] with videoconferencing capabilities.

Of the *synchronous* and a*synchronous telerehabilitation* interventions, clinicians were able to review the participants' performance remotely in all but two studies [22, 45], in which data could not be accessed until the end of the intervention period. There was limited reporting on the software used for data collection, storage and transfer, which often appeared to be integrated within bespoke systems [23, 25, 33]. One study reported using the MySQL relational database management system [19]. Only one study reported using encrypted videoconference software [38], while the remaining studies did not report measures taken to ensure data protection/information security.

## Training and support

**Telerehabilitation delivery.** Clinicians delivered the telerehabilitation in over 80% of studies (26/31). Less than 10% was delivered by researchers, with a combination of clinician and researcher delivery in one study [24].

**Clinician training.** Fewer than 20% of studies delivered by clinicians mentioned clinicians' training (5/26).

**Participant and/or carer training and support.** Just over 50% of studies reported participant and/or carer training details (Table 2). No studies provided direct links to their training materials or manuals. Two studies [20, 31] referred to contacting the corresponding authors for a copy of their manuals, but no response was received at the time of submission.

## Access and costs

**Access to telerehabilitation.** Study eligibility criteria excluded individuals with cognitive impairment (n = 21/31) and communication difficulties (n = 10/31). Two studies reported including additional measures to include participants with cognitive or communication difficulties, such as mailing their questionnaires if they had communication difficulties [42, 44].

The majority of studies appeared to provide study equipment, while just over 20% of studies (n = 5/22, excluding the telephone-only interventions), excluded participants who did not have access and/or the ability to operate a smartphone [32, 48], tablet [37], computer [49], internet access [29]. In one study, if participants could not access teleconferencing software at home, they attended a telehealth centre or received treatment in a separate room at the clinic with no direct contact with the clinician [37].

**Costs of delivering interventions.** Cost analysis was reported in n = 2 studies. A balance gaming intervention costed 44% less to deliver remotely ($835.61) than in-person ($1490.23) [34]. Another study suggested that telephone follow-up had a 68% chance of being cost-saving in terms of health service utilisation, following an estimated a mean cost-saving of £311 in the telerehabilitation group compared with a usual care control [42].

## Usability and acceptability

**Participant-reported outcomes.** Participant-reported outcomes of usability and acceptability were reported in 35% of studies (n = 11/31), and frequently measured using quantitative

measures, of which three studies included validated tools [25, 34, 44], or qualitative feedback (Table 1). In general, studies reported high levels of usability and acceptability, most frequently reported as user satisfaction, with telerehabilitation regardless of the different modes of delivery and platforms. Only one study measured the clinicians' perspective [29], with findings that both participants and therapists were satisfied with the system despite dissatisfaction with some of the system aesthetics and the difficulty of the tasks.

Only one study compared two different modes of telerehabilitation delivery and reported that participants reported higher satisfaction and confidence using videoconference compared with telephone call for assessment of their functional status [32]. When compared with an in-person in-clinic control or usual care, participants reported comparably high levels of satisfaction with videoconference in combination with computer game-based therapy for upper limb rehabilitation [25, 29, 39] and balance training [33]. Participants also reported high levels of usability with telephone review of a computer game-based therapy for balance training [34], compared with in-clinic use of the system. In studies where telerehabilitation provided additional contact or support, such as telephone contact compared with usual care [21, 42] or online support compared with information only [44], participants tended to be more satisfied in the telerehabilitation group.

Participants receiving a multi-component intervention to improve physical activity levels including a mobile app and an in-person group rehabilitation program reported high levels of satisfaction, but many reported preferring the combination of the app and exercise program compared with either the app or exercise program alone [28].

**Adherence.** Only 25% of studies (8/31) reported adherence-related outcomes [19, 25, 26, 28, 29, 32, 42, 49]. Reporting of adherence varied across studies (Table 1); for example, some studies reported the percentage of sessions completed [25, 32, 42], others reported the percentage of participants that completed all the sessions [26] or the average treatment time [19, 29], and not all studies reported adherence for the control groups.

In general, high levels of adherence were observed and, where applicable, adherence was comparable in telerehabilitation and control groups and was consistently at least 80% [25, 32, 49]; although in one study, low compliance with the smartphone app component of a multi-component intervention (50% of participants used the app) was explained by difficulties using the technology [28].

**Adverse events.** Only just over 20% of studies reported on adverse events, with n = 5 having no adverse events and n = 2 having expected adverse events such as upper limb pain [25] and fatigue [25, 26]. Incidence and type of adverse events were comparable in the intervention and control groups.

**Facilitators and barriers to use.** In general, telerehabilitation offered participants increased opportunity for therapy [29, 46]. Positive feedback for telerehabilitation for stroke included improved access to, and interaction with, therapists [19, 48] and participants appreciated the telerehabilitation contact provided [20, 41]. Telephone reviews were more accessible and less disruptive to daily routine than in-person care [31, 42].

Technology-related barriers included: telephone tag [20]; internet connectivity issues [25, 28, 39]; the availability of the technology required [24, 28, 37]; equipment costs [33, 34]; difficulties using the device [28]; the need for additional training or support [24, 45]; and dissatisfaction with the aesthetics of the system [29]. One study added that assistance for technical support was required less frequently as participants progressed through the intervention [25].

Wariness of technology was a barrier to recruitment in one study where n = 21 individuals declined participation due to concerns about use of mobile health technology [32]; although n = 120 participants were recruited, and dropouts were similar (≤20%) in both videoconference and telephone call groups.

## Published protocols of ongoing trials

Study characteristics of included protocols are summarised in Table 3. Table 4 provides a summary of telerehabilitation intervention characteristics from the protocols, with comprehensive descriptions of the interventions to be delivered found in S4 File.

## Telerehabilitation interventions

**Intervention delivery.** The protocols (n = 10) identified suggest that ongoing trials are investigating telerehabilitation interventions targeting physical function (n = 5), upper limb rehabilitation (n = 2) and self-management including health behaviour and secondary prevention (n = 3). Interventions continue to centre around the use of telephone (n = 5) or videoconference (n = 4); one of which uses both telephone and video call in the telerehabilitation group [53]. However, two studies include telerehabilitation contact via an online platform only [51, 56]. Six protocols describe *telesupport* interventions, four protocols describe *asynchronous telerehabilitation* interventions, while none describe *synchronous telerehabilitation*. Most studies are being conducted in Canada (n = 3). See Table 3.

All interventions include a combination of intervention components, such as computer game-based therapy reviewed via videoconference [50, 55] or telephone [58]. Other intervention components that will be used in combination with video or telephone review, such as activity monitors used only by participants to self-monitor their physical activity [52, 57] or upper limb movement [55], and access to online or smartphone-enabled educational content [53, 59], were not reported on here, as they will not be reviewed remotely by the clinician.

Intervention durations ranged from three weeks to six months (Table 4); all the 6-month interventions are categorised as *telecoaching*. Frequency of telerehabilitation contact varies across the protocols from three times/week to contact twice/month.

**Training and support.** Seven protocols describe participant training (Table 4). The level of support anticipated includes additional clinician support in n = 2 protocols, carer support in n = 6 protocols and technical support in n = 5 protocols. Two protocols state they will deliver clinician training [53, 57] (S3 File).

**Access and costs.** As observed in the RCTs, the eligibility criteria have not substantially changed, in that people with communication or cognitive challenges continue to be excluded. Similarly, the majority of studies continue to provide the necessary equipment.

**Usability and acceptability.** Increased numbers of protocols are planning to measure adherence (60%), with all telerehabilitation systems having capability to monitor usage. Measurement of usability and acceptability outcomes remains low (40%) with plans to measure via qualitative feedback [51, 53] and self-developed [53, 58] and validated questionnaires [50, 51, 58].

## Risk of bias of included studies

Risk of bias of included studies is summarised in S5 File. The additional studies (n = 11) showed a similar risk of bias pattern to the studies included in the Cochrane review [13]. Three studies were judged to have high risk of bias [28, 35, 46]. Six studies were judged to be at unclear risk of bias [19, 30, 32, 38, 45, 49]. One study was judged to be at a low risk of bias [42].

## Discussion

To our knowledge, this is the first review to focus on synthesising the evidence for stroke telerehabilitation intervention delivery relating to factors that may influence its uptake and continued use. Given the available evidence related to its effectiveness, this review aimed to

**Table 3. Table of study characteristics—Protocols n = 10.**

| Study ID | Intervention | Control | Outcomes of interest |
|---|---|---|---|
| Country | Including telerehabilitation component and digital and non-digital co-interventions | | Primary clinical outcome and outcomes relevant to usability (adherence, satisfaction) |
| Allegue 2020 | Video-conferenced exergame use for upper limb rehab | Upper limb exercise program with strengthening, range of motion, and functional activities. | Fugl-Meyer Upper Extremity Assessment |
| Canada | | | Adherence; Motivation, using Treatment Self-Regulation Questionnaire-13; Satisfaction, using Modified Short Feedback Questionnaire |
| Blanton 2019 | Constraint-induced movement therapy (CIMT) with carer involvement. Web-based interactive education for the carer while stroke survivor completes home-based CIMT. | Stroke survivor completes home-based CIMT, with no material provided for the carer. | Stroke survivor: Wolf Motor Function Test; Carer: Center for Epidemiologic Studies Depression Scale and Family Caregiver Conflict Scale |
| USA | | | Usability (all carer-related): experience via exit interview, satisfaction feedback forms, Modified Computer Self-Efficacy Scale, Post Study System Usability Questionnaire |
| Chaparro 2018 | Physical activity education and incentive program with weekly telephone calls plus home visit every 3 weeks, and self-monitoring using Sensewear accelerometer and daily subjective physical activity chart. | Usual follow up, including 2 medical reviews at 1- and 6-months post- discharge. | 6-minute Walk Test |
| France | | | |
| Chau 2019 | Multidisciplinary stroke care online platform (including 30 self-care videos) plus video calls with nurse and blood pressure home monitoring device. | Usual stroke rehabilitation services: hospital-based health education, information about local community-based/outpatient rehabilitation services. | Stroke Self-Efficacy Questionnaire & EQ5D-5L |
| Hong Kong | | | Adherence to video call sessions; User Satisfaction Questionnaire; Interviews for user feedback on acceptability, usefulness, difficulties, utility and satisfaction within home setting |
| Chen 2018 | Exercise rehabilitation training and electromyography-triggered neuromuscular stimulation (ETNS) assisted by carer and reviewed via weekly videoconference. | Exercise rehabilitation training and electromyography-triggered neuromuscular stimulation (ETNS) reviewed in-person. | Functional MRI |
| China | | | |
| Gauthier 2017 | Game-based constraint-induced movement therapy (CIMT) in-home with supplemental videoconferencing and telephone contact with therapists plus smartwatch biofeedback. | 1- In-clinic traditional CIMT + home use of mitt | Wolf Motor function Test & Motor Activity Log |
| USA | | 2- In-home gaming CIMT + in-clinic therapist consultation plus smartwatch biofeedback | Adherence to intervention components |
| | | 3- In-clinic standard upper limb rehabilitation | |
| Guillaumier 2019 | Tailored online education program for quality of life and secondary prevention | Usual care plus signposting to generic information available online | EQ5D |
| Australia | | | Adherence |
| Sakakibara 2017 | Telephone lifestyle coaching, self-management manual and self-monitoring kit (health report card, blood pressure monitor, activity monitor and diaries). | Memory training program with same telephone contact as the IG (attention control). | Health Promoting Lifestyle Profile II |
| Canada | | | |
| Sheehy 2018 | At-home virtual reality with rehabilitative exercises for standing balance, stepping, reaching, strengthening and aerobic fitness, delivered via Jintronix and Kinect with remote monitoring and telephone or email contact with therapist. | iPad with apps selected to rehabilitate cognition, hand fine motor skills and visual tracking/scanning and telephone or email contact with therapist (contact same as IG). | Feasibility assessed via uptake, adherence, retention, adverse events, usability and acceptability, including the Physical Activity Enjoyment Scale, and costs |
| Canada | | | |
| Sureshkumar 2018 | Smartphone-enabled, carer-supported, educational intervention with carer support and telephone support from clinician. | Usual care. | Modified Rankin Scale |
| | | | Smartphone app usage monitored |
| India | | | Cost-effectiveness (direct costs of healthcare and rehabilitation and indirect costs to family, travel etc) |

IG–intervention group; CG–control group; EQ-5D –EuroQOL-5D; MRI–Magnetic Resonance Imaging.

synthesise the evidence regarding the practicalities of delivering telerehabilitation in stroke care. There are complex challenges to embedding telerehabilitation into healthcare services and also generating transferable knowledge about scaling up and routinising this service model [60]. This work is essential as it informs the efforts needed to maintain stroke rehabilitation services in the era of Covid-19 and beyond.

## Telerehabilitation intervention delivery

The studies included in this review mainly used videoconferencing or the telephone, either alone or in combination with other intervention components, to deliver stroke telerehabilitation. This review also suggests that that telerehabilitation most often sits as a component within a complex intervention, and its centrality to the intervention and the level of interaction with clinical teams vary. One of the challenges in the synthesis of our findings was the variations in how telerehabilitation was delivered and how to categorise these approaches. We used three categories or definitions of telerehabilitation e.g. synchronous and asynchronous telerehabilitation and telesupport. We searched four international physiotherapy organisations' resources, and were unable to find standardised definitions [61–64], this echoes previous findings that there appears to be no universally agreed definition of telerehabilitation [10]. Use of standardised terminology to describe telerehabilitation delivery may help clinicians identify the most appropriate ways to include telerehabilitation within their services, find the most appropriate resources to support set-up of such services, and communicate effectively with patients about how their telerehabilitation will be delivered. This may improve both patient and clinician understanding and satisfaction, and could be addressed by bringing the community together to agree international consensus, for example in a Delphi study [65].

Given the rise in the popularity of more advanced technologies to deliver telerehabilitation for stroke, it is interesting to note that in terms of modes of delivery, the telephone has not been overlooked as a platform, due to its accessibility and simplicity of use. Alternatively, an advantage of videoconference identified by this review was its potential to facilitate synchronous telerehabilitation whereby participants received real-time clinician interaction and review of their rehabilitation. The only study that compared two telerehabilitation modes of delivery [32], found higher participant satisfaction and confidence with videoconference compared with telephone. Advice given for stroke care during the COVID-19 pandemic is that videoconference is superior to telephone, but telephone consultation is superior to no consultation [66]. Nonetheless, although the current COVID-19 pandemic has accelerated efforts to overcome technological challenges associated with telerehabilitation technologies, improved access to the required hardware and internet connection, and appropriate training and support are required for the implementation of videoconference clinically.

## Training and support

Provision of appropriate training and ongoing support, for both the patients and clinicians, is essential to improve confidence and levels of usability and acceptability contributing to the adoption and maintenance of telerehabilitation in clinical practice [67, 68]. Reporting on training of people with stroke was good (>50% studies) but there was a notable omission of clinician training (20% studies). This is needed to ensure the sustainability of telehealth interventions in the health service and address the urgent and increasing need for stroke rehabilitation worldwide [2]. Additionally, limited detail was available on the content or delivery of the training. As telehealth continues to be used in the response to the pandemic, we are presented with an opportunity for shared learning. Access to these training materials and resources may have facilitated the rapid set-up of telehealth services required during the

**Table 4. Table of TR characteristics—Protocols.**

| Study ID | TT/TR/TC | Telerehabilitation intervention components | Dose | | | | | Training | | | Additional support | | |
|---|---|---|---|---|---|---|---|---|---|---|---|---|---|
| | | | Intervention duration | Number of sessions | Session duration | Other | Reported | Setting | Dose | Training description | Clinician | Carer | Technical |
| **Physical function (n = 5)** | | | | | | | | | | | | | |
| Blanton 2019 | TS | Online interactive platform for carers (remote monitoring) | 4–6 weeks | 6 modules | NR | CIMT for stroke survivor | ✓ | NR | NR | Instructed in use. | - | ✓ | - |
| Chaparro 2018 | TS | Telephone; Accelerometer (self-monitor walking) | 6 months | 1 / week call | NR | - | NR | - | - | - | - | - | - |
| Chen 2018 | AT | Videoconference | 3 months | 1 / week video | NR | Therapy 65 mins x 5/ week | NR | - | - | - | - | ✓ | - |
| Sheehy 2018 | AT | VR/ gaming therapy (remote monitoring); Telephone or email contact | 6 weeks | Therapy 5/ week + 1 / week contact | Therapy 30 mins | - | ✓ | In-person 2 inpatient + 1 at home | 3 sessions of 45–60 mins | Training on system and games, including troubleshooting, manual. | ✓ | ✓ | ✓ |
| Sureshkumar 2018 | TS | Telephone; Smartphone enabled videos (not TR) | 6 weeks | 1 / week call | NR | Smartphone app use no specific dose | ✓ | NR | 45–60 min | Training with practice including competency check | - | ✓ | ✓ |
| **Upper limb rehabilitation (n = 2)** | | | | | | | | | | | | | |
| Allegue 2020 | AT | Videoconference; VR/ gaming therapy (remote monitoring) | 8 weeks | Video 3/ week for 2 weeks, 2/ week for 2 weeks, 1/ week for 4 weeks. | NR | Therapy 30 mins x 5 / week | ✓ | In-person at home | 30-minute | Training session with technician. | - | - | - |
| Gauthier 2017 | AT | Videoconference; VR/ gaming therapy (remote monitoring); Accelerometer (self-monitor upper limb movement) | 3 weeks | Therapy 3 / day for 10 days, 6 video | 1.5 hours / day | 4 in-person consults | ✓ | Outpatient | NR | Education on use of technology | - | - | - |
| **Self-management (n = 3)—Self-efficacy (n = 1), secondary prevention and HRQoL (n = 1), control of risk factors (n = 1)** | | | | | | | | | | | | | |
| Chau 2019 | TS | Videoconference; Online educational platform accessed via study tablet (not TR) | 6 months | Video 1/ month, phone 1/ month | 30–45 min video call | - | ✓ | | | Instructed how to use device kit | ✓ | ✓ | - |
| Guillaumier 2019 | TS | Online platform; Physiological sensors | 12 weeks | 1 / week | NR | Fortnightly prompts | ✓ | Self-led | NR | Letter/email detailing how to access the program | - | ✓ | ✓ |

*(Continued)*

**Table 4.** (Continued)

| Study ID | TT/ TR/ TC | Telerehabilitation intervention components | Dose | | | | Reported | Training | | | Additional support | | |
|---|---|---|---|---|---|---|---|---|---|---|---|---|---|
| | | | Intervention duration | Number of sessions | Session duration | Other | | Setting | Dose | Training description | Clinician | Carer | Technical |
| Sakakibara 2017 | TS | Telephone<br><br>Self-monitoring kit includes pedometer | 6 months | 2 / month in month 1 then 1 / month plus 5 check in calls | 30–60 min calls, 5–10 min check in calls | - | NR | - | - | - | - | - | - |

TR–telerehabilitation; AT–asynchronous telerehabilitation; ST–synchronous telerehabilitation; TS–tele-support; NR–not reported; Min–minutes; H–hours; VR–virtual reality.

current COVID-19 pandemic and in the future. As well as sharing of training resources to support healthcare providers who are introducing remote service delivery and research teams investigating this service model, there may be value in documenting individual experience and wider context learning to prevent duplication and open discussion about optimal delivery. Establishing and sharing resources such as Standard Operating Procedures (SOPs), trouble shooting documents, creating contingency plans, training webinars and manuals would allow for the generation of a knowledge base. This would assist in guiding the processes around telerehabilitation, and may help overcome challenges of embedding such interventions in clinical practice [69].

## Access and costs

In response to the pandemic, and even after the threat of COVID-19 subsides, the "new normal" for rehabilitation services is likely to include a certain amount of telerehabilitation [70–72]. With this shift, it is important that digital equality disparities regarding restricted access and/or low digital literacy are not exacerbated. Those without access to, or who could not use technology were not eligible to participate in over 20% of the studies in the current review. Additionally, study participants were frequently provided with the required equipment, which may not be feasible in practice. There is the risk of delivering less and lower-quality care to the most underserved, by allowing internet access and device ownership to become social determinants of health [60, 73, 74]. Efforts must be made to address the specific needs of those with low digital literacy so that the use of digital technologies does not risk excluding and further disadvantaging this population [75, 76]. Solutions to encourage equitable access may include: additional in-person appointments for those without access to the required equipment; flexibility of clinicians to use the equipment available to the patients, redistribution of refurbished devices; and, providing education, training and support to encourage digital literacy [74, 77].

The reported cost analyses suggest that telerehabilitation may provide a cost-effective alternative that may enable delivery of rehabilitation superior to usual care without the time or resource required for in-person rehabilitation. However, the reporting of intervention delivery costs (2/31 studies; 4/10 protocols) is insufficient to inform service providers implementing telerehabilitation in stroke care, and the lack of clear and consistent reporting of the methods used with insufficient detail to replicate. The need for improved reporting on cost-effectiveness of telerehabilitation has previously been reported [78, 79]. The lack of cost information available is surprising given that cost is often cited as a barrier to setting up telerehabilitation, but equally could be deemed cost-saving and efficient in the long run. Evaluation of cost-effectiveness comparing telerehabilitation and usual care costs, considering start-up costs, clinician time, travel and healthcare utilisation, should be prioritised and incorporated into future telerehabilitation research in a real-world context [13, 78, 79].

## Usability and acceptability

Research on the usability and acceptability of telerehabilitation is essential to enhance uptake and sustainability of service delivery [16, 80, 81]. While these outcomes continue to be inconsistently reported, the findings of this review were encouraging in relation to usability and acceptability. Participant-reported usability and acceptability with telerehabilitation (reported in 11/31 studies of which three used validated outcome measures) was high across various platforms, comparable to in-person interaction and superior to inactive control arms, such as usual care. Additionally, the high levels of adherence to telerehabilitation interventions observed were comparable to in-person rehabilitation, and no safety concerns related to the delivery of telerehabilitation interventions were reported. While adherence in the completed

trials was poorly (25% papers) and inconsistently reported, a greater proportion of the protocols for ongoing trials stated they will measure adherence (60%). Perceived benefits of telerehabilitation reported included improved accessibility to clinicians, increased opportunity for rehabilitation and time efficiency for both patient and clinician. This is particularly pertinent at present given the challenges for both patients accessing and service providers delivering rehabilitation due to the COVID-19 pandemic [REF]. Barriers reported were frequently related to ease of use, which was influenced by the difficulty of using the system as well as connectivity and technical issues. This resulted in low compliance with a smartphone app component of one intervention due to difficulty using it, and frequent requirement of additional support. To ensure successful translation of telerehabilitation into stroke management, it is necessary to ensure the benefits of telerehabilitation are not outweighed by technical challenges. Telerehabilitation service delivery should include easy to follow guidelines for use that are tailored to users' functional abilities and preferences, the opportunity to trial or practice telerehabilitation during a familiarisation period, availability of ongoing technical support, and options for in-person appointments if required. Additionally, consistent reporting of usability- and acceptability-related outcomes throughout the pathway from research to practice, including the use of validated outcome measures and exploration of the factors influencing of usability and acceptability through patient and public involvement (PPI) and qualitative feedback, would facilitate optimisation of interventions to improve uptake and ongoing engagement with telerehabilitation in stroke care.

## Limitations

To make the review process more efficient, we included studies from a Cochrane review with our database searches identifying additional papers published since its last search date [13]. It is also possible we missed relevant studies which were not published in English. Additionally, by including only published protocols, not all trial registrations, it is likely that there is additional ongoing research in this area. For efficiency, data extraction and risk of bias assessment were carried out by a single reviewer with verification by a second reviewer rather than by two independent reviewers. Additionally, all the studies included in this review were conducted prior to the pandemic outbreak. Therefore, it is likely that due to the rapid deployment of various digital devices to deliver contact-free care, that other methods to deliver telerehabilitation have and will continue to emerge across the literature.

## Conclusion

Given the available evidence, the aim of this systematic review was not to provide definitive conclusions regarding the effectiveness of telerehabilitation, but to search and synthesise the evidence regarding the practicalities of integrating telerehabilitation in stroke care. The main findings of this review are that stroke telerehabilitation is generally usable and acceptable and can work well with appropriate training and technical infrastructure. This review recommends improved shared learning from service users and providers to optimise rehabilitation outcomes, patient experience and service quality. It highlighted the need for standardised terminology to describe telerehabilitation, potentially through a Delphi study. As telerehabilitation continues to be used in the response to the COVID-19 pandemic, this review highlighted the need for consistent reporting of the practicalities, challenges and costs of implementation of telerehabilitation into stroke care. This systematic review provides a deeper understanding of telerehabilitation delivery and its translation into stroke practice.

## Supporting information

**S1 File. MEDLINE search strategy.**
(DOCX)

**S2 File. Data extraction form.**
(DOCX)

**S3 File. Description of telerehabilitation interventions (papers).**
(DOCX)

**S4 File. Description of telerehabilitation interventions (protocols).**
(DOCX)

**S5 File. Risk of bias summary and graph.**
(DOCX)

**S1 Checklist. PRISMA 2009 checklist.**
(DOCX)

## Author Contributions

**Conceptualization:** Aoife Stephenson, Sarah Howes, Katy Pedlow, Suzanne M. McDonough.

**Data curation:** Aoife Stephenson, Sarah Howes, Paul J. Murphy.

**Formal analysis:** Aoife Stephenson, Sarah Howes.

**Investigation:** Aoife Stephenson.

**Methodology:** Aoife Stephenson, Sarah Howes, Paul J. Murphy, Katy Pedlow, Suzanne M. McDonough.

**Project administration:** Aoife Stephenson, Sarah Howes.

**Supervision:** Judith E. Deutsch, Maria Stokes, Katy Pedlow, Suzanne M. McDonough.

**Validation:** Katy Pedlow, Suzanne M. McDonough.

**Writing – original draft:** Aoife Stephenson, Sarah Howes.

**Writing – review & editing:** Aoife Stephenson, Sarah Howes, Paul J. Murphy, Judith E. Deutsch, Maria Stokes, Katy Pedlow, Suzanne M. McDonough.

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
