## [Decision Letter · Decision Letter 0]

10 Aug 2021

PONE-D-21-13527

Factors influencing the delivery of telerehabilitation for stroke: a systematic review

PLOS ONE

Dear Dr. Howes,

Thank you for submitting your manuscript to PLOS ONE. After careful consideration, we feel that it has merit but does not fully meet PLOS ONE’s publication criteria as it currently stands. Therefore, we invite you to submit a revised version of the manuscript that addresses the points raised during the review process.

PRISMA guidelines are guidelines are for reporting, not for conduct; please revise your mention of PRISMA accordingly.

We look forward to receiving your revised manuscript.

Kind regards,

Lisa Susan Wieland

Academic Editor

PLOS ONE

Journal Requirements:

Reviewers' comments:

Reviewer's Responses to Questions

**Comments to the Author**

1. Is the manuscript technically sound, and do the data support the conclusions?

Reviewer #1: Yes

Reviewer #2: Partly

2. Has the statistical analysis been performed appropriately and rigorously? 

Reviewer #1: Yes

Reviewer #2: Yes

3. Have the authors made all data underlying the findings in their manuscript fully available?

Reviewer #1: Yes

Reviewer #2: Yes

4. Is the manuscript presented in an intelligible fashion and written in standard English?

Reviewer #1: Yes

Reviewer #2: Yes

5. Review Comments to the Author

Reviewer #1: This is a very clearly presented, well written and comprehensive review with useful information.

I have no suggestions or feedback (although found one typo - line 451 on page 39 says REF instead of listing actual reference).

Reviewer #2: Thank you very much for the opportunity to review this article. It addresses a very important topic in the area of stroke rehabilitation and I believe it will be of great interest to stroke clinicians and researchers alike.

Introduction

Line 68: You cite previous reviews regarding the effectiveness of telerehabilitation for stroke. Please consider including a brief summary sentence regarding this evidence to provide context for the audience regarding this evidence (and its current limitations).

Methods

Searches were limited to systematic reviews or protocols of systematic reviews. Your research aims looked at exploring facilitators and barriers to use, usability and acceptability. Arguably, data relating to your outcomes of interest would be found in qualitative, or mixed methods studies, please provide justification as to why these were not included in the review, and the limitations that may result from this omission.

Lines 112-114: Please provide references for the software used.

Line 135: Please include a copy of your customised form

Line 146: Please address the limitations of only 20% of the data extraction being checked

Please provide dates of searches carried out so readers can understand the currency of the results.

Results:

Line 271: Only one study compared two different modes of telerehabilitation delivery and reported that 272 participants reported higher satisfaction and confidence using videoconference compared with telephone call for assessment of their functional status. Given that telerehabilitation is important in delivering rehabilitation for stroke during the pandemic and beyond, what are the implications for future research and recommendation regarding study designs such as this (comparing different telerehab interventions)? Consider addressing in the discussion

Line 286: only one quarter of studies reported adherence – limitations related to conclusions regarding adherence should be addressed

Facilitators and barriers to use were reported however these again are taken from a limited number of studies- how were these derived, were these facilitators and barriers included in the results section of the papers, if not what are the limitations of this?

Line 300: In general, telerehabilitation offered participants increased opportunity for therapy29,46 ensure you are not overstating this result as only 2 papers are cited to support this.

Risk of bias: The methods section states the studies will be deemed high risk of bias if judged as high risk of bias in any domain. On review of the supplementary material it appears to me that approximately 12 studies have a high risk of bias in at least one domain. There are also many other where the risk is unclear. Could you please clarify your assessment of risk of bias and the impact where the risk is unclear. What impact does this assessment have on the results of your review?

Discussion

Technical requirements- given age of the studies what are the implications of changes technology and privacy requirements

Impact on results and conclusions of high rates of exclusion for participants with cognitive or communication deficits

Line 397: You make a great point regarding lack of reporting on training. Consider going beyond recommendations for this specifically and looking at overall reporting of these studies which may be strengthened using reporting checklists such as the Tidier checklists which recommend detailed reporting of such things as therapist training.

Line 416: You address impact of exclusion of participants without access to technology, also discussed is people’s hesitancy to participate due to technology – what are the implications for this on results on adherence, usability, satisfaction?

You address need for better economic evaluation, please address quality/rigor of economic evaluations provided in this review.

Line 445: no safety concerns related to the delivery of telerehabilitation interventions were reported. However, less than a quarter of studies reported adverse events- please ensure you are not overstating this conclusion.

Line 456 to 458: contains recommendations for telerehabilitation service delivery- please cite evidence supporting this or clarify who is making these recommendations and what this is based on.

Limitations:

Stroke telerehabilitation is a very broad category and includes different types of interventions aimed at different types of impairments e.g., physical vs cognitive vs communication. Consider acknowledging these differences and the impact this may have on the outcomes you have been addressing in this review.

Conclusion:

Line 479: The main findings 480 of this review are that stroke telerehabilitation is generally usable and acceptable and can work well 481 with appropriate training and technical infrastructure. However, a significant percentage of studies in the review did not report acceptability and usability- please ensure you are not overstating this conclusion.

6. PLOS authors have the option to publish the peer review history of their article (what does this mean?). If published, this will include your full peer review and any attached files.

Reviewer #1: **Yes: **Kate Laver

Reviewer #2: No

---

## [Author Response · Author response to Decision Letter 0]

24 Sep 2021

Thank you for reviewing this manuscript. Our response to reviewer comments has been uploaded. Please note that the submission portal would not approve submission without the original manuscript and the revised manuscript. The Revised Manuscript and Response to Reviewers are at the end of the PDF document.

---

## [Decision Letter · Decision Letter 1]

9 Mar 2022

Factors influencing the delivery of telerehabilitation for stroke: a systematic review

PONE-D-21-13527R1

Dear Dr. Howes,

We’re pleased to inform you that your manuscript has been judged scientifically suitable for publication and will be formally accepted for publication once it meets all outstanding technical requirements.

Kind regards,

Amir-Homayoun Javadi, PhD

Academic Editor

PLOS ONE

Additional Editor Comments (optional):

Reviewers' comments:

Reviewer's Responses to Questions

**Comments to the Author**

1. If the authors have adequately addressed your comments raised in a previous round of review and you feel that this manuscript is now acceptable for publication, you may indicate that here to bypass the “Comments to the Author” section, enter your conflict of interest statement in the “Confidential to Editor” section, and submit your "Accept" recommendation.

Reviewer #1: All comments have been addressed

2. Is the manuscript technically sound, and do the data support the conclusions?

Reviewer #1: Yes

3. Has the statistical analysis been performed appropriately and rigorously? 

Reviewer #1: Yes

4. Have the authors made all data underlying the findings in their manuscript fully available?

Reviewer #1: Yes

5. Is the manuscript presented in an intelligible fashion and written in standard English?

Reviewer #1: Yes

6. Review Comments to the Author

Reviewer #1: No additional feedback

7. PLOS authors have the option to publish the peer review history of their article (what does this mean?). If published, this will include your full peer review and any attached files.

Reviewer #1: No

---

## [Editor Report · Acceptance letter]

15 Mar 2022

PONE-D-21-13527R1 

Factors influencing the delivery of telerehabilitation for stroke: a systematic review 

Dear Dr. Howes:

I'm pleased to inform you that your manuscript has been deemed suitable for publication in PLOS ONE. Congratulations! Your manuscript is now with our production department. 

Kind regards, 

on behalf of

Dr. Amir-Homayoun Javadi 

Academic Editor

PLOS ONE